# Pain and Its Management in Patients Referred to a Geriatric Outpatient Clinic

**DOI:** 10.3390/jpm13091366

**Published:** 2023-09-08

**Authors:** Krzysztof Rutkowski, Mateusz Wyszatycki, Krystian Ejdys, Natalia Maria Hawryluk, Małgorzata Stompór

**Affiliations:** 1The Nicolaus Copernicus Municipal Polyclinical Hospital in Olsztyn, 10-045 Olsztyn, Poland; 2Marie Sklodowska-Curie Specialist Hospital in Zgierz, 95-100 Zgierz, Poland; 3Students’ Scientific Group, Department of Cardiology, Medical University of Bialystok, 15-089 Białystok, Poland; 4Department of Family Medicine and Infectious Diseases, Medical Faculty, Collegium Medicum, University of Warmia and Mazury, 10-719 Olsztyn, Poland

**Keywords:** pain management, pain, geriatric outpatient clinic, elderly

## Abstract

(1) Background: A major problem affecting geriatric patients is pain. In addition to pain, a significant problem of old age is dementia and depression, which can hinder the diagnosis and treatment of pain. The aim of this study was to analyse the prevalence of pain in patients treated in a geriatric outpatient clinic and the treatment used. (2) Methods: The analysis was based on the records of 937 patients who visited the Geriatric Outpatient Clinic in Dobre Miasto between 2015 and 2020. Based on records containing data dating back to their first visit to the hospital, patients’ experiences of pain, the presence of depressive symptoms and dementia, and the pharmacological treatment used for pain (analgesics and coanalgesics) were analysed. (3) Results: Pain complaints were reported by 311 patients (33.2% of the study group), 76% of the complaints were from females. The mean age of the patients was 78 years (SD = 8.45). At least one analgesic drug was taken by 107 patients (34.4%). The most commonly used analgesics were opioids (63 patients, 58.87%), especially tramadol. Of the potential coanalgesics, the largest number of patients used an antidepressant. (4) Conclusions: Despite the widespread prevalence of pain among the elderly, only about one-third of them were taking pain medication, mainly in the form of weak opioids. Patients with symptoms of dementia were found to report pain less frequently.

## 1. Introduction

One of the medical challenges we face in the 21st century in developed countries is the increasing number of patients over 60 years of age [1,2,3]. Multimorbidity, typical of an ageing population, is associated with a risk of pain in patients. In 2020, the International Association for the Study of Pain (IASP) modified the definition of pain, which is now characterised as “an unpleasant sensory and emotional experience associated with or resembling actual or potential tissue damage” [4]. Specifically, chronic pain, from which many elderly people suffer, is defined as “pain that lasts or recurs for longer than 3 months” and can be either primary or secondary to another disease [5]. It is estimated that, globally, one in five adults suffer from some sort of chronic pain symptom. This concerns between 25 and 50% of elderly patients, reaching up to 83% in elderly patients in nursing houses [6,7,8]. The main causes of pain in the geriatric population include cancer, neuropathic pain, musculoskeletal changes, and chronic post-traumatic or postsurgical pain [9]. Pain occurrence is related to a significant drop in quality of life, disrupting the daily activity of the elderly as well as their sleep quality [10,11,12]. In Poland, according to the national senior citizens’ health survey PolSenior 2, chronic pain (lasting more than 3 months) was reported by 47.6% of people aged over 60 years [13]. Chronic pain is particularly troublesome for elderly patients due to its multifactorial aetiology and difficulties in causal treatment, as well as its negative impact on seniors’ physical and psychological functioning [14]. This figure may be underestimated, especially given the common neuropathic pain in the elderly population, whose symptoms such as itching, tingling, or numbness may not be classified as pain complaints at all [15].

Basically, quality of life decreases with age, so elderly age is correlated with less social and leisure activities, household activity engagement, and a higher risk of disability. Quality of life in the elderly population further decreases with the presence of pain symptoms [10,16,17]. Studies have reported that pain mainly interferes with one’s overall lifestyle, including their daily activities, mood, and walking [12]. Nonetheless, it is worth mentioning the “well-being paradox”, which states that, while older people deal with cognitive and physical disruptions and other loss experiences, including a higher risk of disability, surprisingly, their well-being is not necessarily significantly decreased compared to younger individuals [18]. In a recent study comparing different age groups from a tertiary care pain centre, it was found that, compared to younger age groups, people over the age of 64 have a greater tolerance threshold for pain ailments, as well as a lower rate of suffering from depression. Moreover, a negative correlation between pain self-efficacy and disability has been found to be a characteristic of elderly patients [19].

The correct diagnosis of pain is also a challenge in elderly patients due to the prevalence of cognitive impairment in this group, and with it the inability to correctly describe pain and its intensity, location, or nature. Communication problems among people with dementia, as well as failure to recognise non-specific manifestations of pain, lead to the neglect of treatment in people with cognitive impairment. Often the ‘disease mask’ of pain in patients with dementia may be the severity or fluctuating course of behavioural disorders [20]. Among other major geriatric problems, chronic pain often accompanies or even causes depression [21]. In dementia, pain has been reported as a factor that causes changes in patient behaviour [22,23].

Chronic pain is also often a cause of disability in older people. The PolSenior 2 found that chronic pain was more often reported by people who were incapacitated in activities of daily living, had limited mobility, and used mobility aids. A patient’s disability in older age may also be one of the obstacles to diagnosing the cause of pain due to difficulties in accessing tests or specialist consultations [24].

Difficulties also exist in the management of pain in older people. This is due to the presence of physiological changes associated with ageing, chronic diseases, and concomitant polypharmacotherapy, which can make the response to the analgesic treatment used difficult to predict, often resulting in over-caution in the use of effective analgesics, e.g., opioids [25,26,27]. Many analyses of the type of pain treatment used in Poland have been carried out, and it has been shown that not all patients reporting pain receive adequate analgesic treatment [28,29]. Furthermore, the aforementioned disability of patients with chronic pain significantly limits the use of alternative pain treatments such as rehabilitation. 

The aim of this study was to assess the prevalence of chronic pain complaints among patients who were referred to a geriatric outpatient clinic in relation to the cognitive functioning of patients and reported symptoms of mood disorders. The analgesic treatment used, which, so far, appears to have been inadequate in this group of patients, was also assessed.

## 2. Materials and Methods

### 2.1. Participants

Our study was retrospective in nature, and the materials for analysis were collected from the records of 937 patients over 60 years of age who first visited the Geriatric Outpatient Clinic of the Complex of Health Care Institutions in Dobre Miasto (Warmińsko-Mazurskie Voivodeship) between 2015 and 2020.

### 2.2. Study Procedure

During their first visit, the reason for seeking medical attention was noted in the patients’ records. Of the possible reasons, patients whose main reason for seeing the geriatrician was pain were selected for further analysis, and data on the medications used, both prescription and non-prescription, were collected in the records. Additionally, patients who exhibited symptoms of depression and cognitive decline were screened for further analysis. The study was approved by an ethics committee headed by Tomasz Stompór MD, Ph.D.

### 2.3. Depression Symptoms Assessment

Information was collected on the intensity of depressive symptoms, as assessed by the Geriatric Depression Scale (GDS-4) and the Geriatric Depression Scale (GDS-15). Patients who scored 1 out of 4 on the GDS-4 or >5 points on the GDS-15 were classified as possibly depressed.

### 2.4. Cognitive Screening

Patients’ cognitive functioning was assessed using the following scales: MMSE (Mini Mental State Examination, a short scale for assessing cognitive function according to Folstein), CDT (Freund Clock Drawing Test, according to Freund), and AMTS (Abbreviated Mental Test Score, according to Hodgkinson—Polish language version according to Skalska). Patients with MMSE scores below 23, CDT ≤ 4 points, and AMTS ≤ 6 points were considered to have probable dementia.

### 2.5. Pharmacotherapy Analysis

The analgesics and co-analgesics taken by patients were also analysed by being divided into drug groups [30,31]. For the purposes of qualitative and quantitative analyses, all medicines were coded based on the Anatomical Therapeutic Chemical classification system recommended by The World Health Organization, and their numbers were calculated.

### 2.6. Statistical Analysis

Licensed TIBCO Statistica software [Version 14.0.0.15, TIBCO Software Inc., Kraków, Poland] was used to statistically analyse the results. In this paper, all results are expressed as the mean (SD) and median (range). To evaluate the normality of distribution of the variables, the Shapiro–Wilk test was applied. As most of the investigated variables were not normally distributed, the nonparametric Mann–Whitney test was employed. The Shapiro–Wilk test was applied to evaluate the normal distribution of the demographic data. Spearman’s rank correlation analysis and a chi-square test were used to assess the relationships between parameters. The significance level was set to *p* < 0.05.

## 3. Results

Of the 937 patients (708 women and 229 men) who visited the Geriatric Outpatient Clinic, pain was reported by 311 patients (33.2%), and for all of which, pain was the main reason for their visit.

There were 237 women (76.2%) and 73 men (23.5%) in the group of patients who cited pain as the main reason for consulting the geriatrician. The mean age of the patients seeking medical attention for pain was 78 years (SD = 8.45). The relationship between age and pain experience among patients was examined, and it was found that younger patients complained of pain more frequently (r_s_ = −0.08, *p* < 0.05).

Pain medication was taken by 147 patients (15.7%, for *n* = 937). At least one analgesic medication was taken by 107 of the 311 patients who cited experiencing pain as the main reason for their geriatric consultation (34.4%). It was found that patients seeking help for pain were more likely to take an analgesic than those reporting to a geriatrician for other reasons (r_s_ = 0.22, *p* < 0.05). Opioids were the most commonly used analgesics, with 63 patients (58.9%) using them for pain treatment; tramadol was the most commonly used opioid (54 patients). Strong opioids were used by only seven patients. Non-steroidal anti-inflammatory drugs (NSAIDs) were taken by 54 patients (50.5%), as shown in Table 1. A total of 14 patients (4.5%) were taking an opioid and a non-steroidal anti-inflammatory drug at the same time. Overall, the most commonly used analgesics were tramadol (54 patients, 38 of whom were taking a combination treatment consisting of tramadol and paracetamol), paracetamol (52 patients, as above), and diclofenac (17 patients). Many patients were taking more than one analgesic at the same time, with a maximum of three drugs (Figure 1).

The percentage of pain patients taking one or more drugs with a potential coanalgesic effect is presented in Figure 2. A total of 48 patients (15.4 per cent, for *n* = 311) were also taking drugs that potentially acted as coanalgesics, such as antiepileptic drugs, selective serotonin reuptake inhibitors (SSRIs), tricyclic antidepressants, quadricyclic antidepressants, and glucocorticosteroids (Table 2). Of the potential coanalgesics, antidepressants were most commonly used (24 patients), and the most commonly used form of which was SSRIs. Of this group of drugs, escitalopram and sertraline were most commonly prescribed. Of the steroids, prednisone was the most commonly used drug, and valproic acid was the most commonly used antiepileptic drug.

The correlation between the paired variables serum creatinine concentration or eGFR and pain medication intake at the time of enrolment was examined. No statistically significant correlations were obtained.

The relationship between patients’ age and reporting to the doctor for pain complaints was investigated, and it was found that younger patients were more likely to report pain complaints (r_s_ = −0.08; *p* < 0.05). No correlation was obtained between patient age and medication groups.

Among the 937 patients analysed, dementia affected 151 patients (16.24%). There were 35 patients with dementia (11.3%) who complained of pain. The relationship between the variable of citing pain as a reason for reporting and CDT, AMTS, MMSE test scores was examined, and it was found that patients with higher test scores were more likely to report pain than patients with dementia (especially advanced dementia) (*p* < 0.05) (Table 3). The relationship between cognitive functioning and pain medication use was also investigated; no statistically significant differences were found between patients with or without dementia and pain medication use.

For the 222 patients experiencing pain, the severity of depressive symptoms was assessed using the GDS-4 or GDS-15 test, with 149 of these patients (67.1%) receiving a score indicating the possibility of co-occurring depression. Depressive symptoms were not found to be more prevalent in patients with pain compared to patients without pain.

## 4. Discussion

In a retrospective analysis, it was found that, in about one-third of patients, the main reason for visiting the Geriatric Outpatient Clinic was pain. In the PolSenior 2 study, chronic pain was reported by 47.6% of patients in those aged 65 years old and older [13]. In a study conducted in Finland, two out of five patients reported seeing a primary care physician specifically because of pain [32]. Epidemiological studies on chronic pain among the elderly worldwide have estimated the prevalence of pain to be 55% in people between 60 and 75 years of age and up to 62–85% in people over 75 years of age [9,33]. In a study conducted in India, among the general population, pain occurrence was reported by 40.4% of people between the ages of 60–74 and 11.3% people of aged 75 and above, and pain complaints were mainly made by the female representatives of the population [10]. It has also been shown that women are more likely to report pain than men, but such a relationship was not confirmed in the material presented here [9,13,32].

Despite this widespread prevalence of pain among the elderly, the analysis presented here found that only about one-third of patients reporting pain take analgesics. This may be due to a number of reasons, including doctors’ fear of potentially harmful side effects after taking the drugs, such as kidney damage or gastrointestinal bleeding, dependence on opioid drugs, fear of worsening patients’ cognitive functioning, an increased number of falls, and/or patients’ own fears of potentially harmful side effects or feeling unwell while taking the drugs [34,35,36]. Using available patient-reported outcomes, the present analysis investigated the association of renal function heights with analgesic intake, but did not confirm the effect of finding renal insufficiency on prescribing frequency.

However, even when treatment is implemented, it is not always effective. Based on the results of the present study, it was found that patients seeking medical attention for pain were more likely to take an analgesic than others; however, experiencing pain remained the most important reason for hospital visits among the elderly, and pain control was not satisfactory. Evaluating the efficacy of the drug groups used, especially in chronic pain, is difficult due to the multifactorial nature of the complaints, the complexity of the pain syndromes, and the difficulty in designing long-term studies [37]. In short-term pain management, the efficacy of individual preparations can be readily demonstrated, as reflected in the nationally published recommendations for pain management in the elderly, but long-term management emphasises the role of topical treatment and non-pharmacological management, especially rehabilitation [38]. Indirectly, the ineffectiveness of pain management by physicians in Poland is also evidenced by the fact that a two-fold increase in sales of over-the-counter drugs, including analgesics, has been observed over the past eight years. In a 2019 study by Weiner et al. on the use of OTC painkillers, 83.6% of older people purchased such drugs. The most commonly taken appeared to be ibuprofen (65.91%) and paracetamol (54%) [28]. In our study, paracetamol was the most commonly used OTC drug, followed by diclofenac.

Comparing the results obtained with similar results obtained in Poland, in a study by Podczaska et al. conducted in 2016 among 392 nursing home residents aged between 75 and 102 years, without dividing them into those experiencing pain or not, pain medication was taken by 113 patients (28.8%), but only 84 patients claimed they were taking medication regularly. Of these, 53 patients (63%) were taking non-opioid analgesics, of which 51 patients were taking NSAIDs, two patients were taking paracetamol alone, nine patients were taking tramadol in monotherapy, two patients were taking tramadol with paracetamol, and one patient was taking morphine [21]. Compared to this study, there was a significantly higher prevalence of opioid use in the data presented. In a study conducted in Italian geriatric departments, only 49% of patients with pain symptoms received any treatment. Patients were treated mainly with NSAIDs. Strikingly, 74.5% of patients receiving treatment declared that their treatment was ineffective or not effective enough [17].

In another study by this author from 2018 (also conducted among nursing home patients but with an emphasis on those with probable dementia), out of 96 patients with pain, only 33 people (34%) were taking analgesics. Overall, 13 people (14%) were taking non-opioid analgesics, of which 3 were taking NSAIDs, and 10 were taking paracetamol. A total of seven people (7%) were taking tramadol with paracetamol, and two people (2%) were taking buprenorphine. A total of 11 people (11%) were taking a non-opioid drug in concert with an opioid. A total of 20 (60%) of the 33 people taking medication were using an opioid [39]. Compared to the study described above, the present study found a similar proportion of people taking an opioid drug (58.9%), while significantly more people were taking NSAIDs (50.5% of patients in our analysis). These differences are probably due to a different cross-section of patients (patients in care homes receive medication prescribed by a doctor, whereas the study group included outpatients who may have also purchased over-the-counter medication).

In studies conducted in other European countries, significant differences in the consumption of analgesics have been observed. In a 2019 observational study focusing on a group of 94, 820 elderly patients in Northern Italy, non-steroidal anti-inflammatory drugs were taken by 36.62% of patients, while opioids were taken by 12.48% of patients [40]. Meanwhile, in France, in a group of patients over 55 years of age receiving analgesic treatment, the prevalence of using weak opioids ranged between 23 and 27%, and strong opioid usage ranged between 1.4 and 4.8%, while a much smaller number of patients were taking NSAIDs (between 3.0 and 10.8%), with paracetamol being the most commonly used form of this drug class [41].

In our study, we also analysed the potential coanalgesic medications taken by patients presenting with pain. During their first visit to the Geriatric Outpatient Clinic, 44.85% of patients taking an analgesic reported that they were also taking a coanalgesic. However, it is not known whether these were prescribed for pain or for another reason. In the PolSenior 2 study, an analgesic together with a coanalgesic was reportedly taken by 7.5% of patients. Interestingly, despite inadequate pain control, no patients were chronically treated with serotonin-norepinephrine reuptake inhibitors (SNRIs), which are recommended as the standard of care for neuropathic pain, while some patients were taking antiepileptic drugs. 

In epidemiological studies, the prevalence of depression in people over 60 years of age ranges from between 4.3% and 63% [9]. Moreover, depression often accompanies the experience of chronic pain, often as part of so-called ‘total pain’ [15]. An Italian study on nursing homes showed that depressive symptoms occurred in 62.8% of the studied group, of which 77.3% had a mild behaviour disorder. Moreover, a statistically significant correlation between pain and depressive symptoms was found, as was their link to patient irritability or lability and aggression [22]. However, in our analysis, no correlation was found between the presence of pain and depression. Also, no significant statistical correlations were found between taking pain medication and the severity of depressive symptoms. It is worth mentioning that the data concentrated on the pathophysiology of pain and depression (suggesting that these two conditions have a similar pathophysiology) and the reason why people suffering from pain are likely to develop depression and vice versa. This relationship has been labelled by some authors as “pain syndrome depression” or “depression pain dyed” [9,42,43]. As such, conducting sufficient pain control can be useful as a preventative tool against depression.

In our analysis, dementia affected 151 people with pain (16.24%). Among the patients reporting pain, there were 35 people with dementia (11.3%), and it was also shown that those with more severe cognitive impairment reported pain less frequently than those with less severe dementia. Interestingly, a study by Neumann-Podczaska et al. found that patients with dementia were less likely to receive analgesic treatment compared to patients without dementia [21]. A similar correlation was obtained in the above study. An Italian study conducted on nursing homes patients revealed a dementia prevalence level of 77%. Of these, 46.4% patients reported pain symptoms, but a limitation mentioned by authors of this study was that only 42.5% of all patients were able to provide a reliable answer. Surprisingly, 97% of patients with dementia and coexisting pain were prescribed analgesics [22]. There are data that emphasize that, while dementia leads to inappropriate, unexpected behaviour, pain symptoms may also be accompanied by changes in the patient’s behaviour, including depression, aggression, and agitation [22,23]. It is important to be aware of the coexistence of pain and changes in behaviour to develop better tools for pain diagnosis among patients with dementia while bearing in mind that dementia is a great geriatric problem.

The relationship between age and pain perception among patients was examined, and it was found that pain was the most frequently reported reason among younger patients (r_s_ = −0.08, *p* < 0.05). In the PolSenior 2 study, pain complaints were reported most frequently by patients between 85 and 89 years of age. An explanation for these differences may be that, in the study presented in this paper, many patients exhibited cognitive problems, which may have hindered proper communication in reporting pain complaints [44].

## 5. Conclusions

Diagnosing pain in elderly patients remains a challenge not only due to the multi-morbidity of the elderly but also the difficulty of contact with the patient. Adequate pain treatment should be provided so that older people can maintain the highest possible quality of daily functioning. Bearing in mind that quality of life drops with every decade and even quicker with the coexistence of pain, it is crucial to provide patients with proper pain treatment.

## Figures and Tables

**Figure 1 jpm-13-01366-f001:**
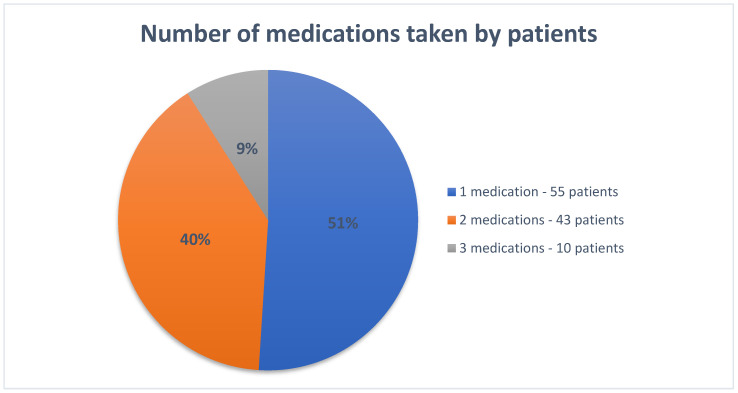
Percentage of patients with pain complaints taking one or more pain medications based on the records of patients reporting pain complaints.

**Figure 2 jpm-13-01366-f002:**
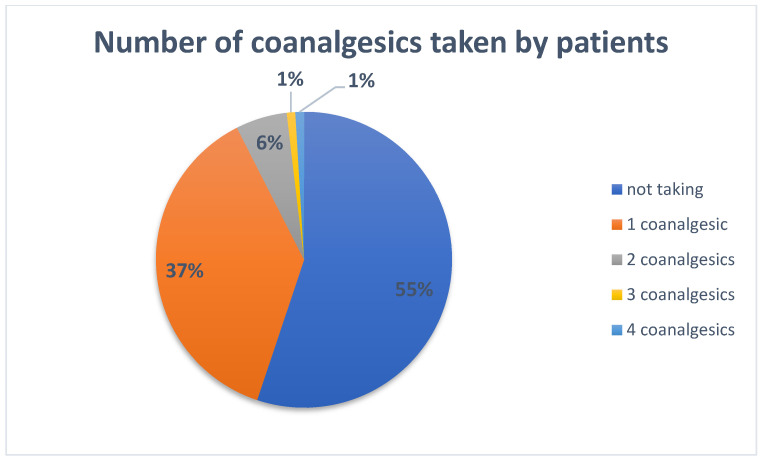
Percentage of pain patients taking one or more drugs with a potential coanalgesic effect.

**Table 1 jpm-13-01366-t001:** Analgesics taken by pain patients.

Name of Drug	Number of Patients	%* (*n* = 107)
Paracetamol	14	13.1%
Paracetamol with tramadol	38	35.5%
Paracetamol (in monotherapy and in combination with with tramadol)	52	48.6%
Weak opioids	56	52.3
Tramadol (in monotherapy and in combination with with paracetamol)	54	50.5
Tramadol (in monotherapy)	16	15%
codeine	2	1.9
Potent opioids	7	6.5
buprenorphine	4	3.7
oxycodone	2	1.9
fentanyl	1	0.9
Non-steroidal anti-inflammatory drugs	54	50.5
diclofenac	17	15.9
ketoprofen	8	7.5
nimesulide	7	6.5
naproxen	7	6.5
ibuprofen	4	3.7
meloxicam	4	3.7
aceclofenac	4	3.7
celecoxib	2	1.9
lornoxicam	1	0.9

* The percentages do not add up to 100% due to the fact that some patients were taking more than one pain medication.

**Table 2 jpm-13-01366-t002:** Drugs with potential coanalgesic effects taken by patients with pain complaints.

Coanalgesic Group	Number of Patients	% (*n* = 311)
Antidepressants	24	7.8%
Escitalopram	8	2.6%
Sertraline	7	2.3%
Mianserin	6	2%
Mirtazapine	2	0.6%
Amitriptyline	1	0.3%
Anti-epileptic	20	6.4%
valproic acid	7	2.3%
Carbamazepine	4	1.3%
Levetiracetam	2	0.6%
Gabapentin	2	0.6%
Oxcarbazepine	2	0.6%
Pregabalin	2	0.6%
Lamotrigine	1	0.3%
Glucocorticosteroids	15	4.8%
Prednisone	9	2.9%
Methylprednisolone	5	1.6%
Hydrocortisone	1	0.3%

Pain medication intake and renal function parameters.

**Table 3 jpm-13-01366-t003:** Results of tests assessing cognitive function in patients reporting pain.

Type of Test	Number of Patients *	Average Score	Probable Dementia	Correlation r_s_: Pain as Reason for Report vs. Test Score (*p* < 0.05)
CDT	199	5.1 +/− 1.74	65	−0.09
AMTS	69	8.2 +/− 2.3	15	−0.2
MMSE	61	23.7 +/− 5.34	20	−0.15

* The number of patients does not add up to 311 because some patients had more than one type of test to assess cognitive function.

## Data Availability

Not applicable.

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
