# Peer review of "Pain and Its Management in Patients Referred to a Geriatric Outpatient Clinic"

_jpm, 2023, doi:10.3390/jpm13091366_

Round 1
Reviewer 1 Report
Population aging is a worldwide situation that requires social, economic, and health improvements.
This paper shows the prevalence of chronic pain in elderlies (around 33%) from a city in Poland. According to previous reports in other countries, these data demonstrate that chronic pain is a debilitating condition requiring our attention to find alternatives to relieve the pain of this fast-growing sector of the population.
I suggest that authors improve graphics quality since they are poor resolution.
Author Response
Dear Reviewer,
I am writing to sincerely thank for your review of our article entitled „Pain and its management in patients referred to a geriatric outpatient clinic” by Krzysztof Rutkowski, Mateusz Wyszatycki, Krystian Ejdys, Natalia Hawryluk, MaÅ‚gorzata Stompór. Also, I would like to reply to your questions and suggestions:
- We clarified that the study design was retrospective and that the patients were not assessed later on prospectively.
- We have added the information about ethical committee approval.
- We have tried to insert more graphically clear images to our work
All changes and edits are highlighted.
I hope that I have managed to provide a relevant answer to your review.
Sincerely,
Natalia Hawryluk
Reviewer 2 Report
The manuscript “Pain and its management in patients referred to a geriatric outpatient clinic” describes the prevalence of pain in geriatric patients , in relation to the patients’ cognitive functioning. While it does not provide any novelties, it is a well written article and adds to the evidence of pain presence in the elderly and problematics related to pain reporting and treatment in this group of patients.
I have some comments which I would like the authors to address:
1. in the Material and Methods section:
· Is the study design retrospective? If yes, state so explicitly . It is not clear to me whether patients reporting pain were assessed later on prospectively.
· In the text in, the study procedure section page 2 , you state that patients with pain were selected for further analysis; what does this imply specifically?
· In the study procedures please explain if depression and cognitive screening were performed for all patients. Is it normal clinical practice for patients in your center to fill in the questionnaires mentioned or was the collection performed prospectively?
· Has the study been approved by an ethics committee? I failed to find any statements in regards.
2. in the Results section:
· I would suggest changing the subtitle “Taking painkillers” to something like “Drug prescription” if these painkillers were prescribed by the physicians
· I would suggest removing Figure 1 and 2 and just describe the information in the text.
· In page, sentence 147, you mention the correlation between serum creatinine and eGFR to pain medication, however, you never mention before, neither in the aim nor in the method section that you were going to also assess this. And it is unclear to me what kind of correlation did you study and why
· I would change both subtitles 3.5 and 3.6 since in the present form it is not clear what you are assessing, in fact, I would suggest removing all subtitles in the results section and just keep separate paragraphs for the descriptive data and correlation analysis.
Overall is good.
Author Response
Dear Reviewer,
I am writing to sincerely thank for your review of our article entitled „Pain and its management in patients referred to a geriatric outpatient clinic” by Krzysztof Rutkowski, Mateusz Wyszatycki, Krystian Ejdys, Natalia Hawryluk, MaÅ‚gorzata Stompór. Also, I would like to reply to your questions and suggestions:
Ad.1
- We clarified that the study design was retrospective and that the patients were not assessed later on prospectively.
- The material for the analysis was collected from the records of patients who attended their first visit to the Geriatric Outpatient Clinic of the Complex of Health Care Institutions in Dobre Miasto (Warmińsko-Mazurskie Voivodeship) between 2015 and 2020. Then we analyzed data only of those patients whose reason to visit the Geriatric Outpatient Clinic was pain.
- Depression and cognitive screening were performed only for patients who reported relevant symptoms during the medical examination.
- We have added the information about ethical committee approval.
Ad.2
- We have been looking for the correlation between eGFR and drugs taken by patients, including painkillers, coanalgesics and anti-depressants. We have found no statistically significant correlation.
- We have tried to insert more graphically clear images to our work
- We removed the subtitles, as you suggested.
All changes and edits are highlighted.
I hope that I have managed to provide a relevant answer to your review.
Sincerely,
Natalia Hawryluk
Round 2
Reviewer 2 Report
The authors have answered to my previous concerns.